# Liver Trauma: Management in the Emergency Setting and Medico-Legal Implications

**DOI:** 10.3390/diagnostics12061456

**Published:** 2022-06-13

**Authors:** Angela Saviano, Veronica Ojetti, Christian Zanza, Francesco Franceschi, Yaroslava Longhitano, Ermelinda Martuscelli, Aniello Maiese, Gianpietro Volonnino, Giuseppe Bertozzi, Michela Ferrara, Raffaele La Russa

**Affiliations:** 1Department of Emergency Medicine, Policlinico Gemelli/IRCCS University of Catholic of Sacred Heart, 00168 Rome, Italy; saviange@libero.it (A.S.); veronica.ojetti@unicatt.it (V.O.); christian.zanza@live.it (C.Z.); francesco.franceschi@unicatt.it (F.F.); 2Foundation “Ospedale Alba-Bra Onlus”—Department of Emergency Medicine, Anesthesia and Critical Care Medicine, Michele and Pietro Ferrero Hospital, 12060 Verduno, Italy; lon.yaro@gmail.com; 3Research Training Innovation Infrastructure, Research and Innovation Department, Azienda Ospedaliera SS Antonio e Biagio e Cesare Arrigo, 15121 Alessandria, Italy; 4Department of Emergency Medicine-Section of Anesthesia and Critical Care, San Giacomo Hospital-Novi Ligure, 15067 Novi Ligure, Italy; emartuscelli@aslal.it; 5Department of Surgical Pathology, Medical, Molecular and Critical Area, Institute of Legal Medicine, University of Pisa, 56126 Pisa, Italy; aniello.maiese@unipi.it; 6Department of Anatomical, Histological, Forensic and Orthopedic Sciences, Sapienza University of Rome, 00185 Rome, Italy; gianpietro.volonnino@uniroma1.it (G.V.); michelaferrara13@gmail.com (M.F.); 7Department of Clinical and Experimental Medicine, University of Foggia, 71100 Foggia, Italy; raffaele.larussa@unifg.it

**Keywords:** liver trauma, blunt abdominal trauma, emergency department, bleeding, WSES

## Abstract

Traumatic abdominal injuries are life-threatening emergencies frequently seen in the Emergency Department (ED). The most common is liver trauma, which accounts for approximately 5% of all ED admissions for trauma. The management of blunt liver trauma has evolved significantly over the past few decades and, according to the injury’s severity, it may require massive resuscitation, radiological procedures, endoscopy, or surgery. Patients admitted to the ED with blunt abdominal trauma require a multidisciplinary evaluation, including emergency physicians, surgeons, radiologists, and anesthetists, who must promptly identify the extent of the injury to prevent serious complications. In case of a patient’s death, the execution of a forensic examination carried out with a multidisciplinary approach (radiological, macroscopic, and histological) is essential to understand the cause of death and to correlate the extent of the injuries to the possibility of survival to be able to manage any medico-legal disputes. This manuscript aims to collect the most up-to-date evidence regarding the management of hepatic trauma in the emergency room and to explore radiological findings and medico-legal implications.

## 1. Introduction

Liver injuries account for approximately 5% of all ED admissions for trauma [1]. Blunt or penetrating abdominal trauma are frequently responsible for significant hepatic lacerations or hematomas, with a high mortality risk due to the uncontrolled hemorrhage. The mortality rate is estimated to be between 10–15% in patients with severe hepatic trauma [1]. The liver is easily vulnerable to trauma for various reasons, including its size, position in the anterior portion of the abdominal region, and fragility of the parenchyma and Glisson’s capsule. In younger populations, this vulnerability is increased due to the greater weakness of the connective tissue and the greater flexibility of the ribs [2]. Liver vascularization consists of a dual arterial circulation and a portal venous supply with large vessels made up of a thin wall, more susceptible to post-traumatic bleeding. According to the Couinaud classification, the liver is divided into eight segments, each one functionally independent from the other as they have their own vascularization and biliary tract. Segment I corresponds to the caudate lobe, segments II (superior) and III (inferior) are placed medial to the falciform ligament, segment IV is placed laterally to the falciform ligament, while the remaining segments constitute the right portion of the liver (V inferomedial, VI inferolateral, VII superolateral, and VIII superomedial) [3].

In over 80% of cases, the posterior segments of the liver are affected by injuries, due to crushing against the bony structures of the trunk (ribs, vertebral column). Furthermore, the right portion of the parenchyma is often involved, due to its size and proximity to the ribs [2]. Left lobe injuries are uncommon and generally due to direct thoraco-abdominal compression, as occurs in road accidents. They are associated, in general, with lesions of the pancreas, heart, duodenum, and transverse colon [4]. The most frequent etiologies include motor vehicle crashes, pedestrian accidents, and traumatic falls [1]. Other causes include work-related accidents, sports accidents, gunshots, and blunt and penetrating wounds. The American Association for the Surgery of Trauma (AAST) graded liver trauma through the Hepatic Injury Scoring Scale [3,5], which recognizes VI grades of trauma, from minor lesions (which represent approximately 80–90% of hepatic lesions) to those that are fatal [6]. Other issues considered in the scoring system are computed tomography findings (imaging criteria), operative criteria based on the injured area, and pathologic criteria. The emergency physician must perform the primary and the secondary survey, a laboratory evaluation, and radiological imaging to define the degree of lesions and to identify the most appropriate treatment (operative or non-operative). Moreover, the treatment of liver trauma patients involved in motor vehicle crashes, workplace accidents, or caused by third individuals may be subject to legal action. For these reasons, the emergency physician must be aware of the medico-legal risk and the importance of correctly keeping healthcare documentation.

## 2. Mechanism of Liver Injury

Blunt liver trauma can be caused by strong accelerations and subsequent decelerations following an impact with an external body or by crush injuries [6]. These mechanisms are most common after motor vehicle collisions, and they are generally associated with rib fractures, pneumothorax, and renal injuries [4].

Acceleration-deceleration injuries are due to the interaction between external forces and ligamentous constraints. Acceleration injuries are due to the liver moving along the coronal plane. Since the right lateral ligament fixes the VII hepatic segment, while the V and VIII segments are mobile, lacerations between the anterior and posterior portions of the parenchyma occur. An additional modality of acceleration injury occurs when acceleration pushes back the liver, with possible laceration of the main hepatic veins or the right hepatic lobe, which are not fixed by ligaments. Deceleration moves the liver against the anterior or posterior chest wall due to enormous high-speed forces that suddenly stop, resulting in injury to the anterior or posterior segments [7].

Crush injury makes the liver collide between the anterior and the posterior chest wall, with potential laceration due to the compression force [8]. In the case of a penetrating hepatic injury, the severity of lesions depends on the trajectory and on the firing distance of the tool or missile, and liver injuries can range from parenchymal to vascular laceration (minor or major). In addition, chest wall trauma can be associated with significant liver injury [9,10]. In infants, liver injury can occur during childbirth in a breech presentation [11] or from childhood abuse [12].

Frequently, liver lesions are found on the diaphragmatic face of the liver and can range from subcapsular lesions to complete parenchyma transection. Sporadically, there may be only internal lacerations. The World Society of Emergency Surgery (WSES) classified liver injuries (Table 1) as minor (grade I), moderate (grade II), and severe (grade III-IV) [13]. However, the most frequently used classification of hepatic injury is the Liver Injury Scale of the AAST, updated in 2018, which classifies injuries as follows [14]:

## 3. Patient Evaluation in the Emergency Department

The emergency physician must conduct a primary survey to rule out life-threatening conditions, checking the airway, breathing, and circulation for hypovolemic shock due to liver hemorrhage, neurological status, and environmental conditions [15]. Hypotension and tachycardia may indicate abdominal bleeding; pain or tenderness in the right upper abdominal quadrant may suggest liver injury; and abdominal distension may imply hemoperitoneum. Patients with liver trauma should receive laboratory tests with complete blood counts, hemoglobin, platelets, coagulation parameters, transaminases, liver function tests, creatinine, glucose, and lactate blood levels. In addition, a radiological assessment is fundamental with an initial fast ultrasound examination (FAST) to detect intra-abdominal blood collections [16]. Blood detection in the abdominal cavity can guide the assessment of the trauma severity and the management of the liver injury. Hemodynamically stable patients required a computed tomography (CT) scan with intravenous (IV) contrast medium. CT can identify traumatic liver injury and quantify the abdominal hemoperitoneum [1,17], while immediate access to the operating room is mandatory in hemodynamically unstable patients with a positive FAST examination.

## 4. Radiological Findings

The method to diagnose liver trauma depends on patient hemodynamic status [13,18]. FAST rapidly detects abdominal free fluids but sometimes with suboptimal views, with poor quality images because of, for example, the patient’s body habitus [13,19]. It examines five regions (pericardial, right and left upper quadrant—focusing on hepato-renal and splenic-renal areas—pelvic or suprapubic, and anterior thorax) to detect accumulation of free fluid or blood [20,21].

A CT examination is currently considered the “gold standard” imaging technique in trauma [13]. The CT imaging description of liver trauma includes major findings as hematomas, lacerations, active bleeding, and those that are minor such as periportal vein low attenuation and flat inferior vena cava. Moreover, a CT scan can assess hepatic abscesses, fluid collections, haemobilia, biliary complications, and peritonitis [17]. CT findings can also include the early or the late development of post-traumatic hepatic artery pseudoaneurysm [22]. The detection rate of hepatic injury in patients with blunt trauma is approximately 25% [17]. The mortality rate of patients with liver trauma is approximately 4–12% [17,23,24]. In blunt liver trauma, lacerations appear as irregular low attenuation areas, linear or branching; they can be deep (>3 cm) or superficial (<3 cm) [8,25]. If they are extended in the posterior region of the VII hepatic segment, they can be responsible for retroperitoneal hematoma. Hematoma could also be subcapsular or intraparenchymal. Acute hematomas appear as a hyper attenuation signal (40–60 U) compared to normal liver parenchyma. Active bleeding is a high focal attenuation (mean 150 U) in areas with extravasated contrast fluid. Other “acute” radiological findings include low periportal attenuation parallel to the portal vein and flat inferior vena cava due to hypovolemia. As regards delayed post-traumatic complications, delayed bleeding is the most frequent after blunt hepatic injury in approximately 3–5% of patients with an overall mortality of 18%. Hepatic and perihepatic abscesses appear as areas of fluid attenuation, with gas bubbles or air-fluid levels in the traumatized liver parenchyma. Patients can suffer from abdominal pain and tenderness, fever, and leukocytosis. Blunt trauma can be responsible also for hepatic artery pseudo-aneurysm that appears as a round focal lesion with high attenuation; they have a risk of rupture with hemorrhage. Hepatic lacerations can be responsible for biliary leakage, fistula, biloma, and biliary peritonitis [3,17].

## 5. Management of Liver Trauma in the ED: Focus on Non-Operative Management (NOM)

The management of blunt liver trauma has changed significantly in the last decades [18]. Most severe hepatic injuries are currently treated without surgery. In addition, the use of interventional radiology with vascular embolization procedures has improved liver bleeding control in polytrauma patients [23,26]. Patients admitted to the ED with blunt abdominal trauma require a prompt surgical evaluation. Patients with minor hepatic trauma are treated non-operatively (without surgical intervention). In contrast, hemodynamically unstable patients with ongoing hepatic bleeding often require angioembolization or surgery [18] in addition to fluid and blood transfusions. WSES guidelines recommend choosing a non-operative treatment for hemodynamically stable patients with minor-moderate liver lesions without other internal damage requiring a surgical approach. In case of severe lesions, non-operative management can be possible if continuous monitoring of patients and access to angiography procedures can be ensured. Patients with a CT scan of acute bleeding have to be managed by a skilled surgical team in the operating room.

Emergency physicians have to perform continuous clinical monitoring, blood tests for hemoglobin, and CT scans to control the hemorrhage. Moreover, the management strategy has to include fluid support, blood transfusions, thromboprophylaxis (based on LMWH), enteral feeding (if not contraindicated), and early mobilization in stable patients [13].

The best evidence suggested that LMWH-thromboprophylaxis is required due to the hypercoagulation state of the patient within 48 h of the traumatic event and it is safe in NOM [27]. It is estimated that more than 50% of patients without thromboprophylaxis may develop deep vein thrombosis and pulmonary embolism with high-risk mortality [13]. On the contrary, patients on anticoagulant therapy may require reversal agents to reduce the risk of bleeding [10]. Guidelines recommend oral nutrition after 24–48 h from the hepatic traumatic event in stable patients. In patients admitted to the Intensive Care Unit (ICU), enteral feeding is recommended within the first 72 h [13]. In case of complications, such as bowel obstruction, bowel ischemia, fistulas, gastrointestinal bleeding, and abscesses, it has to be delayed [13]. Most liver lesions heal in approximately three to four months, and a strict long-term follow-up is not indicated [13]. In addition, physical activity can be started again after this period of rest. After the discharge, the emergency physician has to instruct the patient to immediately come back to the ED in the event of increasing abdominal pain, vomiting, or nausea.

Currently, there are no evidence-based guidelines that define the duration of abstention from physical activity and the intensity with which to restart after a liver injury; therefore, any evaluation is limited to a literature review, which appears poor. Healing of a simple liver tear and subcapsular hematoma occurs in 2–4 months, while complex lesions take up to 6 months [28].

On the other hand, the choice of follow-up imaging after this type of injury is controversial. Although CT documentation of resolution was once the standard of care, a follow-up CT is no longer recommended unless clinically indicated. However, some recommend returning to unlimited activity only after a normal CT scan, usually 3 to 6 months after the injury, especially in the case of professional athletes [29].

The management of hepatic trauma should always be multidisciplinary. During the first contact with a health facility, NOM is the first choice for the emergency physicians who admit the patient, although it requires prompt evaluation by the trauma surgeon to provide the best treatment based on clinical conditions and the severity of injuries.

## 6. Management of Blunt Liver Trauma in the ED: Focus on Surgical Treatment

Surgical treatment is necessary to control bleeding in patients with liver trauma who are hemodynamically unstable and do not respond to NOM. This operative management (OM) is a part of the surgical “*damage control*” strategy, and it is needed as soon as possible, together with resuscitation. The procedure of angioembolization or balloon occlusion of the aorta can be considered a definitive procedure in patients with active bleeding or delayed hemorrhage [13].

The choice of surgical technique varies according to the extent of the lesions that have occurred. If there is no evidence of major bleeding at the time of the laparotomy, the following may be considered: electrocautery, topical hemostatic agents, or simple hepatic parenchyma suturing or omental patches [30,31,32,33]. In cases of major bleeding, more invasive procedures should be considered: manual compression and hepatic packing, ligation of vessels, hepatic debridement of necrotic areas, balloon tamponade, shunt or isolation procedures, and hepatic vascular exclusion—techniques that require early intensive intraoperative resuscitation to maintain organ perfusion.

As regards vascular lesions, a distinction between arterial and venous has to be made. If there is a lesion to the hepatic artery, the primary choice should be selective ligation of the hepatic artery, with a selective ligation in the case of isolated lesions alternately on the right or left branches; however, it should always be borne in mind that if the right or common hepatic artery is to be ligated, a cholecystectomy should be associated to avoid gallbladder necrosis, although ligation of the hepatic artery increases the risk of hepatic necrosis, abscesses, and biloma formation. If bleeding persists with such measures, the presence of an accessory hepatic artery should be considered. On the other hand, if the rupture affects the portal vein, these takes priority—keeping the ligation as the last choice and only in cases of hepatic artery integrity—due to the high risk of hepatic necrosis or intestinal mucosal pain, otherwise, liver packing or liver resection should be considered. Liver packing is the least risky method of temporarily addressing severe venous injury.

Complete vascular exclusion without supracoeliac aortic cross-clamping is poorly tolerated in unstable patients [34].

If despite all damage control procedures active bleeding is still present, the use of a resuscitative endovascular balloon occlusion of the aorta (REBOA) and simultaneous exclusion with resuscitative endovascular balloon occlusion of the vena cava (REBOVC) should be considered [35].

Furthermore, in the case of avulsion of the hepatic pedicle, insertion of a veno-venous bypass (porto-caval and cavo-caval) would be the first step to achieve hemodynamic stabilization in order to mitigate the severe detrimental hemodynamic effects caused by inferior vena cava and portal vein occlusion. After this maneuver, an immediate hyperurgent liver transplant can be the only option in a transplant center [13].

Recent studies are focusing on possible hybrid measures that include postoperative angiography-embolization as an additional hemostatic tool in patients who have become hemodynamically stable after initial operative hemostasis or in patients with suspected uncontrolled arterial bleeding despite emergency laparotomy.

## 7. Medico-Legal Implications of Hepatic Trauma

Blunt liver trauma can be responsible for deadly outcomes. Autopsy case studies have shown that lacerations, followed by hematomas [36,37,38,39,40,41,42,43,44], represent the most frequent lesion. Another unusual cause of hemorrhage, which may be associated with delayed death from trauma, is the rupture of a post-traumatic hepatic artery aneurysm [44]. On the other hand, it is believed that a mechanism of origin of the hepatic lacerations is “an extraordinarily high venous pressure that develops in the instant of impact” and they were created experimentally by blunt direct impact to the heart at speeds of 12–18 m/s, as the origin of the pressure gradient [45]. The right lobe appears to be more frequently affected than the others. Nevertheless, liver lesions are frequently associated with lesions to other organs (such as spleen or kidneys) and districts (i.e., rib fractures).

Thus, medico-legal involvement in these circumstances is associated with the possibility that liver injuries can be the subject not only of ascertaining the cause of death in criminal law cases but of complaints about medical liability in civil claims: for example, with earlier access to treatment, the disavowal of even a small injury or a different treatment could determine a salvific outcome [46].

Several factors affect the mortality of patients with hepatic trauma, including liver injury extension, hemodynamic instability, and associated lesions. Notably, the survival rate is inversely proportional to the degree of damage [47].

For patients with grade I-II hepatic trauma associated with other injuries, mortality is approximately 13%, while the mortality rate increases for grade III, IV, V, and VI to 15%, 30%, 65%, and 95%, respectively [48]. Lesions involving the main hepatic veins or the retro-hepatic vena cava are often fatal [49]; additional predictors of mortality in grade IV-V injuries include intraoperative blood loss, hypothermia, acidosis, post-traumatic coagulopathy, and dysrhythmia [50,51].

In this regard, it is important to remember that most deaths during abdominal surgery were intraoperative, with the primary cause of death being exsanguination; instead, multiple organ failure accounted for most of the postoperative deaths [52].

In patients with abdominal injuries, hepatic lesions represent the main cause of death, leading to a deterioration of the patient’s clinical condition. The concomitant thoracic and abdominal injuries result in massive bleeding, with subsequent hypovolemic shock and death. In this type of patient, timely diagnosis and treatment impact survival more than injury severity. An appropriate treatment performed within the first hour of the trauma increases the survival of the polytrauma patient, which conversely decreases if it is performed after an hour and with blood transfusions greater than three liters [53]. Recently, Kim et al. proposed a specific scoring system for determining the prognosis of patients with polytrauma and hepatic trauma (traumatic liver injury scoring system—SSTLI), which improves the ability of a traditional trauma scoring system to predict mortality [54]. The SSTLI employed five clinical measures (total serum bilirubin, prothrombin time, serum creatinine, age, and injury severity score), each of which was scored. A 5-point cutoff was used to divide patients at high risk of developing in-hospital mortality [55].

Furthermore, as regards medico-legal implications, it is crucial to observe that the application of NOM for liver trauma has not been associated with an increased risk of mortality. NOM is estimated to be safe also in patients with grade IV and V liver injuries. Moreover, in severe liver trauma, the therapeutical success rate correlates with an improved damage control strategy [52] that sometimes includes surgery. The spread of whole-body CT in hepatic trauma has progressively reduced the rate of missed injuries in the ED. In addition, the non-surgical procedures of interventional radiology to stop bleeding or bile leaks (such as percutaneous drainages and biliary stenting) resulted in being a safe alternative to laparotomy, with better outcomes [52] and without an increased mortality risk.

On the other hand, adverse circumstances could be classified as complications rather than errors. Therefore, in the event of death due to hepatic trauma, a post-mortem examination is necessary to define the cause of death and to manage the medico-legal litigation [56].

A methodological autopsy examination duly preceded by post-mortem imaging [57,58] cannot prescind from a careful analysis of the macroscopic lesions and the extent of the damage to the liver parenchyma. The forensic pathologist will necessarily have to describe the location and the size of the liver lesions, especially their depth, to correlate the extent of the trauma to the chance of survival [59,60]. The medical examiner’s reports do not always contain these details, which are essential for correlating the extent of the trauma to the chances of survival and establishing whether death is directly linked to the trauma or whether there has been an error in the patient’s management [18].

Furthermore, it is mandatory to sample the lesion to characterize the damage histologically [61,62] (Figure 1). According to a study conducted by Kohlmeier et al., the microscopic examination of hepatic lacerations could also provide information on the time of production of the lesions. Specifically, the presence of neutrophilic infiltration and necrosis of hepatocytes at the site of the laceration is more frequently associated with a survival between 51 min to 7 h and 10 min [63]. Furthermore, albeit according to the common sense, significant fatty liver infiltration is more susceptible to traumatic laceration because it is more rigid, even if in a study by Molina et al. there was no statistical correlation between the degree of fatty change and the presence of liver trauma [64]. Therefore, it will be the overall case by case assessment and the collection of all available evidence that allows for an accurate determination.

The presence of scarring attributable to older liver lesions should lead to suspicion of childhood trauma. Notably, significant hemosiderin deposits were found in Kupffer cells and hepatocytes in chronically abused individuals [65].

## 8. Conclusions

Hepatic trauma is one of the most frequent complications after blunt or penetrating abdominal trauma, representing a life-threatening reason for admission in the emergency room in relation to the extent of the injuries. The management of liver trauma has evolved significantly over the past few decades. To prevent and to avoid serious complications, emergency physicians must promptly recognize the degree of the lesions and choose the most appropriate treatment (surgical or non-surgical). The type of approach (operative or NOM) depends on hemodynamic status, the extent of liver lesions, the presence of associated lesions, and the patient’s comorbidities. The advancement of imaging studies has played an important role in the conservative approach.

The management of these patients must be multidisciplinary, as it requires the involvement of emergency physicians, radiologists, surgeons, and forensic experts to cover all of their needs and to avoid fatal complications.

From a medico-legal point of view, the correct compilation of the medical record represents the keystone to allowing the healthcare workers involved in managing these patients in the ED to prevent potential criminal or civil repercussions. Furthermore, in the event of the patient’s death, the execution of a forensic examination conducted with a multidisciplinary approach (radiological, macroscopic, and histological) is essential to understand the cause of death and to correlate the extent of the injuries to the possibility of survival in order to be able to manage any medico-legal disputes.

## Figures and Tables

**Figure 1 diagnostics-12-01456-f001:**
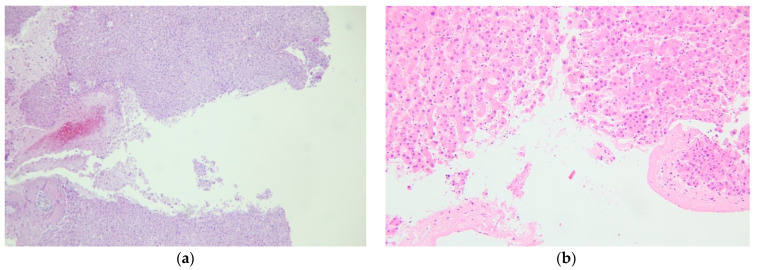
Hematoxylin-eosin staining showing hepatic parenchyma discontinuity and consensual peri-lesional erythrocyte infiltration (**a**) 10× (**b**) 20×.

**Table 1 diagnostics-12-01456-t001:** Liver Injury Scale according to AAST.

Grade	Injury Type	Description
**I**	Hematoma	subcapsular, <10% surface area
	Laceration	capsular tear, <1 cm parenchymal depth
**II**	Hematoma	subcapsular, 10–50% surface area
	Hematoma	intraparenchymal <10 cm diameter
	Laceration	capsular tear 1–3 cm parenchymal depth, <10 cm length
**III**	Hematoma	subcapsular, >50% surface area of ruptured subcapsular or parenchymal hematoma
	Hematoma	intraparenchymal >10 cm
	Laceration	capsular tear >3 cm parenchymal depth
	Vascular	injury with active bleeding contained within liver parenchyma
**IV**	Laceration	parenchymal disruption involving 25–75% hepatic lobe or involving 1–3 Couinaud segments
	Vascular	injury with active bleeding breaching the liver parenchyma into the peritoneum
**V**	Laceration	parenchymal disruption involving >75% of hepatic lobe
	Vascular	juxtahepatic venous injuries (retrohepatic vena cava/central major hepatic veins)
**VI**	Vascular	hepatic avulsion

In grade I–III injuries, mortality is related to the extent of associated injuries, while in high-grade injuries, it depends on the anatomical liver damage.

## Data Availability

No new data were created in this study. All analyzed data were obtained from references listed in the specific paragraph.

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
