# Peer review of "Liver Trauma: Management in the Emergency Setting and Medico-Legal Implications"

_diagnostics, 2022, doi:10.3390/diagnostics12061456_

Round 1
Reviewer 1 Report
Comments to authors
1. There are few grammatical and language errors like in line 3, page 1; line 60 & 89, page 2; line 118 & 164, page 4.
2. FAST is a focused ultrasound examination so 'FAST ultrasound examination' is not a good term (Line 95, page 2).
3. The term 'suboptimal' should be replaced by figures to make it objective (line 96, page 2).
4. The regions to detect free fluid in FAST examination mentioned are incorrect (line 97-98, page 3).
5. The findings in CT scan should also include early or late development of pseudoaneurysm that has management implications.
6. Mechanism of liver injury should precede the section on patient's evaluation in ED.
7. In addition to the trajectory of the bullet, the severity of hepatic injury also depends on the velocity of the missile. This should be included at appropriate place (Paragraph 3, under 'mechanism of liver injuries').
8. The term 'liver traumatic bleeding' should be rephrased (line 160, page 4).
9. The statement 'A laparoscopic exploration is possible when there is doubt about abdominal injury'should be cited elsewhere at appropriate place. Including this statement in operative management of liver injury in hemodynamically unstable patients is misleading. There is no place for laparoscopy in a hemodynamically unstable patient with suspected intra-abdominal hemorrhage.
10. There is repetition of statement ' WSES guidelines suggest the non-operative management of patients 168 with liver trauma in case of hemodynamic stability' in the last paragraph on page 4.
11. The use of variable terms like emergency physicians and surgeons foe managing liver injury is misleading. It is connoting the message that operative intervention should be done by surgeons while physicians should perform the non-operative management. The trauma surgeon should be involved in all aspects of management of a case with liver injury (last paragraph on page 4).
12. The issue of thromboprophylaxis especially using LMWH should be elaborated as it is debatable in hepatic injuries.
13. The last 4 lines under the section 'Management of liver trauma in the ED: focus on non-operative management (NOM)' are incomprehensible and should be revised.
14. The issues of limited, restricted and full physical activity following liver injury must be explained in detail as this is a 'Review article'. The way it is mentioned is ambiguous and incomplete.
15. The timing of follow up after hospital discharge and the way to assess patients should be mentioned. It is a complex issue. There is a lot of debate on the timing, type and choice of radiological procedure for patient assessment during hospital visits. The issue of duration of strict bed rest immediately after the injury and hospital discharge and the means for examining the patient in follow-up should also be clarified (first paragraph page 5).
16. The term 'medical treatment' should be avoided and be replaced by 'non-operative management'. Similarly, the term 'not responding' may be replaced by 'failure of non-operative management' (line 191-202, page 5).
17. Angioembolization may be a definitive procedure in patients with active bleeding or delayed hemorrhage and is not a 'bridging procedure'. This may be corrected appropriately (line 191-202, page 5).
18. The choice of the surgical procedure should be mentioned in detail in different clinical scenarios. Clubbing all possible treatment modalities right from perihepatic packing to liver transplant in a paragraph does not serve the purpose of a review article (line 197-202, page 5).
19. Figure 1 is not matching the description in the legend. Moreover, it is a postmortem image. This should be replaced with a more representative, clinical image.
20. The statement 'The etiology of blunt abdominal trauma with liver injury may require medico-legal examinations' is unclear. All cases of trauma are medicolegal cases (line 207-208, page 5).
21. The term 'blunt heart impact' is unclear (line 213, page 6).
22. The iteration of 'non-traumatic hemorrhage' is superfluous and may be removed.
23. There are only 2 lobes in liver. In this light the statement 'the right lobe appears to be more frequently affected than the others' may be revised.
24. The statement 'liver lesions are frequently associated with lesions to other organs and districts' is unclear (line 217, page 6).
25. Second paragraph on page 6 is unclear and must be rephrased.
26. Paragraph 3-6 on page 6 are not related to directly to medicolegal issue and may be omitted from this section. I would suggest the authors to focus only on practical medicolegal aspects in liver injury in this section.
27. There is repetition of text in the first paragraph of page 7.
28. Lines 270-280 mentions a controversy and then is left to the judgement of the readers. This should be elaborated further by the authors.

Author Response
29th May, 2022
Dear Reviewer,
We want to thank you for your interest in our work and for yourhelpful comments that will greatly improve the manuscript (Manuscript ID: diagnostics-1741330). We appreciate the opportunity to increase the quality of our manuscript.

Reviewer 2 Report
Liver trauma is a life-threatening injury. Correct management, in terms of time, laboratory and instrumental investigations (FAST vs CT) are essential to ensure the best possible outcome for the patient.
This work develops these elements well, outlining the possible management based on the possible scenarios of trauma presentation (stable vs unstable hemodynamics, CT or FAST findings).
Author Response

(The authors gave the same response as above.)

Reviewer 3 Report
The statements in the manuscript are basically to be agreed. However, the technical details of surgical management of blunt liver trauma are extremely general and meaningless in an overview. A more detailed explanation of this would certainly be appropriate. Especially, it would be important to explain damage control surgery of liver injuries in more detail.
Author Response

(The authors gave the same response as above.)

Round 2
Reviewer 1 Report
I would request the authors to change Figure 1 to a clinical one or delete it altogether.
Author Response
Dear Reviewer,
We want to thank you again for your help in improving our work thanks to your comments. We deleted "Figure 1" according to your suggestions.